

# Predicting near-term changes in ocean carbon uptake

Nicole S. Lovenduski[1], Stephen G. Yeager[2], Keith Lindsay[2], and Matthew C. Long[2]

[1]Department of Atmospheric and Oceanic Sciences and Institute of Arctic and Alpine Research, University of Colorado, Boulder, Colorado, USA

[1]Climate and Global Dynamics Laboratory, National Center for Atmospheric Research, Boulder, Colorado, USA

*Correspondence to:* Nicole S. Lovenduski (nicole.lovenduski@colorado.edu)

**Abstract.** Annual to decadal variations in air-sea fluxes of carbon dioxide ($CO_2$) impact the global carbon cycle and climate system, and previous studies suggest that these variations may be predictable in the near-term. Here, we quantify and understand the sources of near-term (annual to decadal) predictability and predictive skill in air-sea $CO_2$ flux on global and regional scales by analyzing output from a novel set of retrospective decadal forecasts of the Earth system. These initialized forecasts exhibit the potential to predict year-to-year variations in the globally-integrated air-sea $CO_2$ flux up to ∼7 years in advance. This initialized predictability exceeds the predictability obtained solely from foreknowledge of variations in external forcing or a simple persistence forecast. The near-term $CO_2$ flux predictability is largely driven by predictability in the surface ocean partial pressure of $CO_2$, which itself is a function of predictability in surface ocean dissolved inorganic carbon and alkalinity. Comparison with an observationally-based product suggests that the initialized forecasts exhibit moderate predictive skill in the tropics and subtropics, but low skill elsewhere. In the subantarctic Southern Ocean and northern North Atlantic, we find long-lasting initialized predictability that beats that derived from uninitialized and persistence forecasts. Our results suggest that year-to-year variations in ocean carbon uptake may be predictable well in advance, and establish a precedent for forecasting air-sea $CO_2$ flux in the near future.

## 1  Introduction

Observations collected over the past few decades indicate that the ocean has absorbed 160 Pg of excess carbon from the atmosphere since the beginning of the industrial revolution (Le Quéré et al., 2018); projections from climate models suggest that ∼540 Pg of excess carbon will reside in the ocean by the end of the century (under the RCP8.5 emission scenario; Ciais and Sabine, 2013). Accurate projections of past and future air-sea $CO_2$ flux are important for quantifying and understanding the changing global carbon cycle and for estimating future global climate change (Le Quéré et al., 2018).

Superimposed on the background of long-term changes in ocean carbon uptake is substantial variability on global and regional scales (McKinley et al., 2017). The recent literature highlights ocean carbon uptake variability that manifests on timescales of years to decades. Interannual variability in globally-integrated air-sea $CO_2$ flux has been estimated to have a standard deviation of 0.31 Pg C yr$^{-1}$ and 0.2 Pg C yr$^{-1}$ from observationally-based products (Rödenbeck et al., 2015) and ocean biogeochemical models (Wanninkhof et al., 2013), respectively, which is on the order of 10% of the global-mean $CO_2$ flux (2.3 Pg C yr$^{-1}$). A global extrapolation of sparse $p CO_2$ observations suggests that there is large variability on decadal



timescales (Landschützer et al., 2016). On regional scales, Southern Ocean studies have highlighted recent air-sea $CO_2$ flux variability on interannual (Wetzel et al., 2005; Lenton and Matear, 2007; Lovenduski et al., 2007, 2013, 2015; Verdy et al., 2007; Wang and Moore, 2012; Hauck et al., 2013; Lenton et al., 2013) and decadal (Fay et al., 2014; Landschützer et al., 2015; Munro et al., 2015) timescales. In the North Atlantic, high air-sea $CO_2$ flux variability has been linked to the North

Atlantic Oscillation (Thomas et al., 2008; Ullman et al., 2009) and the Atlantic Multidecadal Oscillation (Metzl et al., 2010; Breeden and McKinley, 2016), whose spectra peak at interannual and multi-decadal timescales.

Near-term predictions of the climate system (so-called "decadal predictions") are forecasts of climate variability and change on annual, multi-annual, and decadal timescales from global climate models (Meehl et al., 2014). These forecasts are sensitive to both initial conditions (e.g., the atmospheric temperature used to initialize the forecasts) and external forcing (e.g.,

the long-term increase in atmospheric temperature associated with increasing greenhouse gas concentrations; Kirtman et al., 2013). Recent publications highlight near-term predictability and predictive skill in regional surface air temperature, precipitation, Arctic sea ice concentration, oceanic heat content, and the large-scale Atlantic Ocean circulation (Meehl et al., 2009; Robson et al., 2012; Yeager et al., 2012; Meehl et al., 2014; Yeager et al., 2015; Boer et al., 2016; Yeager and Robson, 2017). As prior literature has established a strong link between air-sea $CO_2$ flux and variability in the physical climate system on these

timescales (e.g., McKinley et al., 2017), it follows that air-sea $CO_2$ flux may be predictable in the near-term.

Here, we analyze a novel set of decadal prediction simulations from an Earth System Model (ESM) to investigate near-term predictions of global and regional ocean carbon uptake. On annual to decadal timescales, ESM predictions of the past (so-called "retrospective forecasts") are used to assess both predictability and predictive skill in air-sea $CO_2$ flux. Predictability is the potential to predict the system, based on forecast verification against a model reconstruction. Predictive skill is based

on forecast verification against observations. We further assess the role of external forcing in the predictability of $CO_2$ flux by analyzing a set of uninitialized forecasts run under identical external forcing. By analyzing forecasts of the past, our study establishes a precedent for making skillful predictions of ocean carbon uptake in the near future.

## 2  Community Earth System Model Decadal Prediction System

Our primary numerical tool is the Community Earth System Model Decadal Prediction Large Ensemble (CESM-DPLE;

Yeager et al., in press). In this section, we describe the model and provide details about forecast initialization, ensemble generation, and drift correction. Importantly, we note that this is the first CESM decadal prediction system to include a representation of ocean biogeochemistry. CESM-DPLE uses the same code base as the CESM Large Ensemble (CESM-LE; Kay et al., 2015).

The CESM is a state-of-the-art coupled climate model consisting of atmosphere, ocean, land, and sea ice component models (Hurrell et al., 2013; Danabasoglu et al., 2012; Lawrence et al., 2012; Hunke and Lipscomb, 2008). The ocean physical model

(version 2 of the Parallel Ocean Program; Danabasoglu et al., 2012) has nominal 1° horizontal resolution and 60 vertical levels. The biogeochemical ocean model represents the lower trophic levels of the marine ecosystem (Moore et al., 2004, 2013), full carbonate system thermodynamics (Long et al., 2013), air-sea $CO_2$ fluxes, and a dynamic iron cycle (Doney et al., 2006; Moore and Braucher, 2008).




CESM-DPLE consists of a set of initialized, fully-coupled integrations of CESM that adhere to the protocols for Component A of the Decadal Climate Prediction Project (DCPP), a contribution to the $6^{th}$ Coupled Model Intercomparison Project (Boer et al., 2016). We use the CESM-DPLE system (Yeager et al., in press) that builds on previous CESM decadal prediction efforts (Yeager et al., 2012, 2015) with some modifications (including the addition of ocean biogeochemistry, as noted above).

CESM-DPLE initiates 40 decade long "forecasts" of the Earth system each year from 1954-2015; the start date for each forecast is November 1, in accordance with the DCPP protocols. Each of the model integrations are subject to a common set of historical external forcings (e.g., greenhouse gas concentrations).

The ocean physical and biogeochemical initial conditions for the DP experiments are generated from a forced ocean - sea ice simulation of the CESM. That is, a simulation of the ocean and ice components of the CESM that has been forced with

fluxes computed from the observed atmospheric state over 1948-2015. This simulation is therefore meant to reconstruct the historical evolution of the ocean physical and biogeochemical state over the 1948-2015 period (Figure 1). Hereafter, we refer to this simulation as the "reconstruction". Initial conditions from the atmosphere and land components of the DP experiments are obtained from a $20^{th}$ century simulation of the CESM Large Ensemble (Kay et al., 2015).

Ocean biogeochemistry in the version of the CESM used for CESM-DPLE has been extensively validated in the literature

(Long et al., 2016; Lovenduski et al., 2016; McKinley et al., 2016; Krumhardt et al., 2017; Freeman et al., 2018). In particular, the simulated mean, variability, and trends in surface ocean $pCO_2$ and air-sea $CO_2$ flux from CESM over 1982-2011 compare favorably to estimates from observations for the global average and over most ocean biogeochemical biomes (McKinley et al., 2016; Lovenduski et al., 2016). In Figure 2, we illustrate the comparison between observationally-based estimates of $CO_2$ flux (from the Landschützer et al. (2016) $pCO_2$ product) and estimates produced by the reconstruction over 1982-2015, which

indicates that the model reconstruction does a reasonable job of representing observed spatial patterns (in both magnitude and direction) of the flux across most oceanic regions. The globally-integrated air-sea $CO_2$ flux over 1982-2015 from the observational product and model reconstruction are 1.41 and 1.80 Pg C yr$^{-1}$, respectively (directed into the ocean).

CESM-DPLE initializes an ensemble of 40 simulations each year using round-off level (order $10^{-14}$) perturbations in the initial air temperature field (Figure 1). Each ensemble member is subject to identical external forcing. The number of ensem-

ble members in each forecast ensures statistically robust drift estimates (see below; Boer et al., 2013; Kirtman et al., 2013; Yeager et al., in press).

Following initialization, the coupled model drifts toward its preferred state over the decadal forecast. This is a common problem for full-field initialization decadal prediction experiments (Meehl et al., 2014) and requires a drift correction to be applied to the model forecasts before predictability and predictive skill may be analyzed. We correct the drift by transforming to

anomalies from a drifting climatology, as in Yeager et al. (2012) and Yeager et al. (in press). For a given forecast, $X(L, M, S)$, where L is the forecast length, M is the ensemble member, and S is the start year of the forecast, the drift-corrected forecast anomaly, $X'(L, M, S)$ is defined as

$$X'(L, M, S) = X(L, M, S) - \overline{X(L, M, S)}^{M,S}, \tag{1}$$



where $\overline{X(L,M,S)}^{M,S}$ is the average rate of drift over all forecasts. Note that this method disregards potential dependence of the drift on the external forcing.

Predictive skill in CESM-DPLE may be enabled by external forcing (e.g. the time evolution of atmospheric greenhouse gases) as well as by initialization. To assess the role of initialization in predictability, we compare CESM-DPLE air-sea $CO_2$ flux (generated with the initialization procedure described above) with air-sea $CO_2$ flux from the CESM-LE (McKinley et al., 2016; Lovenduski et al., 2016) over the same historical period. The CESM-LE is a 40-member ensemble of the CESM with fully resolved ocean biogeochemistry that evolves the Earth system from 1920 to 2100 under historical and RCP8.5 forcing (Kay et al., 2015). As such, CESM-LE represents the uninitialized counterpart to the CESM-DPLE system; output from CESM-LE can tell us how the modeled air-sea $CO_2$ flux would evolve over a given decade in the absence of initialization, but under the same external forcing.

## 3 Results

### 3.1 Predictability

Predictability is a property of a system that characterizes the ability for its future evolution to be predicted; this concept is distinct from that of model skill. We quantify predictability by evaluating the ability of the CESM-DPLE initialized forecasts to predict variations in air-sea $CO_2$ flux from the reconstruction. For a given forecast anomaly, $X'(L,M,S)$, predictability is defined as the correlation coefficient of $X'(L,M,S)$ with the corresponding anomaly in the reconstruction; the reconstruction anomaly is obtained by subtracting the climatological mean value over 1955-2015.

The globally-integrated, air-sea $CO_2$ flux anomaly from the initialized CESM-DPLE in forecast year 1 exhibits high correlation with the $CO_2$ flux anomaly from the reconstruction (Figure 3a; r = 0.98). This correlation remains high and statistically significant (at the 95% level, using a two-sided student $t$ test while accounting for autocorrelation in the sample size) for 10 forecast lead years (Figure 3c), suggesting high, long-lasting predictability in the globally-integrated air-sea $CO_2$ flux.

We further investigate whether the predictability in the globally-integrated air-sea $CO_2$ flux is a function of initialization by (1) correlating integrated $CO_2$ flux anomalies from the ensemble mean of the uninitialized CESM-LE simulation with anomalies from the reconstruction, and (2) generating a persistence forecast (autocorrelation as a function of lead time) for the $CO_2$ flux anomalies from the reconstruction. Figures 3a and 3c reveal that the initialization of the forecast does not much improve the prediction from the uninitialized forecast. This is because the strong externally-forced component of the forecast (e.g., the rising $CO_2$ concentration in the atmosphere) provides an important source of predictability in both the initialized and uninitialized forecasts. While the persistence forecast also yields high correlation coefficients, both the initialized and uninitialized forecasts beat persistence for all prediction lead times (Figure 3c).

Figure 3a also reveals interannual variability in the globally-integrated air-sea $CO_2$ flux. While this variability is swamped by the externally forced signal (i.e., the increasing $CO_2$ uptake due to rising atmospheric $CO_2$), we are nevertheless interested in the ability of CESM-DPLE to forecast this year-to-year variability. To accomplish this, we remove the linear trend from





the forecasts and the reconstruction before computing predictability; this method produces estimates of correlation that are not dominated by the trend induced by external forcing. The globally-integrated, detrended, air-sea $CO_2$ flux anomaly from the initialized CESM-DPLE in lead year 1 exhibits high correlation with $CO_2$ flux from the reconstruction (Figure 3b; r = 0.70), suggesting high predictability of ocean carbon uptake variability on interannual timescales, as well. While this predictability drops off with forecast lead time, we nevertheless find high correlations (r > 0.4) between the annual-mean $CO_2$ flux forecast

anomalies and detrended reconstruction anomalies that extend for 7 years, and statistically significant correlations that extend for 10 years (Figure 3d).

Interannual variability in global air-sea $CO_2$ flux may also be affected by interannual variability in external forcing (e.g., volcanoes). As above, we evaluate the role of initialization by calculating uninitialized predictability and estimating persistence. Figure 3 indicates that the initialized forecast exhibits higher predictability than the uninitialized forecast and the persistence

forecast for lead times of 7 and 10 years, respectively. Thus, the CESM-DPLE initialized forecasts have the potential to predict year-to-year variations of globally-integrated air-sea $CO_2$ flux up to 7 years in advance.

The results from our analysis of the globally-integrated air-sea $CO_2$ flux suggest that interannual variations in global ocean carbon uptake may be predictable well in advance. They further indicate that initialization of the forecasts enhances the predictability of future interannual variations over and above the predictability from variations in the external forcing, such as

those imposed by volcanic eruptions, fluctuations in the rate of emissions, or variability in terrestrial $CO_2$ fluxes. This is a particularly meaningful result for those forecasting year-to-year changes in the global carbon budget (e.g., Le Quéré et al., 2018), especially as these forecasting efforts are blind to the externally forced variability in advance (i.e., the external forcing of the future is unknown). In this way, near-term predictions of air-sea $CO_2$ flux variations can help to inform future predictions of land-air $CO_2$ flux and atmospheric $CO_2$.

Given the high predictability and the important role of initialization in forecasts of interannual air-sea $CO_2$ fluxes on a global scale, we next investigate the spatial patterns of air-sea $CO_2$ flux predictability across the global ocean. Here, we use the same statistical techniques as for the global flux, but instead perform analysis in each model grid cell. On a global scale, the evolution of air-sea $CO_2$ flux is dominated by the long-term increase in ocean uptake (see, e.g., Figure 3a), whereas on local and regional scales, the evolution is dominated by interannual variability (Figure 1; see also, e.g., Lovenduski et al., 2016). To

capture the predictability on interannual timescales, we perform analysis on linearly detrended forecasts. Figure 4a illustrates large predictability of initialized $CO_2$ flux across much of the global ocean for forecast lead year 1. The uninitialized forecast (Figure 4b) and the persistence forecast (Figure 4c) indicate lower predictability.

If not external forcing or persistence, what drives the high predictability in air-sea $CO_2$ flux interannual variability? We decompose the predictability of air-sea $CO_2$ flux ($\Phi$) over forecast lead year 1 by considering the predictability of its drivers:

$$\Phi = k \cdot S_0 \cdot (1 - ice) \cdot \Delta pCO_2, \tag{2}$$

where $k$ is the piston velocity (also known as the gas transfer coefficient), $S_0$ is the solubility of $CO_2$ in seawater, $ice$ is the fraction of the ocean covered by sea ice, and $\Delta pCO_2$ is the difference between the oceanic $pCO_2$ and the atmospheric $pCO_2$.



As for $CO_2$ flux, predictability is defined as the anomaly correlation coefficient of each driver variable in forecast year 1 with the corresponding anomaly of that driver variable in the reconstruction, e.g., the correlation of anomalous piston velocities from the forecast with those from the reconstruction. Figure 5 shows the predictability of each of the $CO_2$ flux driver variables,

where the anomaly correlation coefficients are scaled to $CO_2$ flux units (mol m$^{-2}$ yr$^{-1}$) and can be easily compared. The predictability scaling is achieved by multiplying the anomaly correlation coefficient ($r$) by the sensitivity of $CO_2$ flux to each driver variabile ($x$) and the standard deviation of the driver variable timeseries:

$$r \cdot \frac{\partial \Phi}{\partial x} \cdot \sigma_x, \tag{3}$$

where the sensitivities and standard deviations are established from model-estimated quantities in each grid cell (as in, e.g.,

Lovenduski et al., 2007, 2013, 2015), using annual averages from the reconstruction. The $CO_2$ flux predictability is largely driven by predictability in $\Delta pCO_2$ across the global ocean (Figure 5). Our results suggest secondary roles for the piston velocity in the equatorial Pacific, solubility in the North Atlantic subpolar gyre, and sea ice fraction in the Arctic/North Atlantic and high latitude Southern Ocean. Elsewhere, these other driver variables play only minor roles in $CO_2$ flux predictability.

As the large predictability in $\Delta pCO_2$ is caused by predictability of surface ocean $pCO_2$ in our model framework (i.e.,

atmospheric $CO_2$ concentration is prescribed, rather than predicted), we next investigate the drivers of interannual predictability in surface ocean $pCO_2$: dissolved inorganic carbon (DIC), alkalinity (Alk), temperature (T), and salinity (S). We use a similar approach as for $CO_2$ flux, but here the sensitivities are derived from carbonate chemistry approximations (Lovenduski et al., 2007; Doney et al., 2009; Long et al., 2013), and all drivers are scaled to $pCO_2$ units ($\mu$atm) for ease of comparison:

$$r \cdot \frac{\partial pCO_2}{\partial x} \cdot \sigma_x. \tag{4}$$

The surface ocean $pCO_2$, and thus the air-sea $CO_2$ flux predictability for forecast lead year 1 is largely driven by predictability in surface ocean DIC and Alk, with temperature playing a secondary role, and salinity a minor role (Figure 6). The similar predictability of DIC and Alk across many regions hints at an important role for ocean circulation, rather than biological productivity (which has a much larger impact on DIC than Alk), in $CO_2$ flux predictability.

## 3.2 Predictive skill

We next evaluate the predictive skill of the CESM-DPLE forecasts; the skill is a measure of the ability of the forecast to reproduce the observational record. For air-sea $CO_2$ flux, direct observations are rare, and we are constrained to estimates of flux from observations of sparsely sampled surface ocean $pCO_2$. Here, we use as our observational metric the $CO_2$ flux estimated from the Landschützer et al. (2016) surface ocean $pCO_2$ product. This product is a gap-filled estimate of surface ocean $pCO_2$, which, when combined with measurements of atmospheric $pCO_2$, sea surface temperature, salinity, and wind,

yields a monthly estimate of air-sea $CO_2$ flux at 1° x 1° horizontal resolution from 1982-2015 (see also Figure 2a). As the $pCO_2$ observations are rather sparse prior to 1995 (see Figure 2 of Bakker et al., 2016), we calculate skill for the period between 1995 and 2015 only, but show for the interested reader the full observational product timeseries.



The CESM-DPLE initialized predictions exhibit some skill at representing the globally-integrated air-sea $CO_2$ flux in forecast lead year 1 (Figure 3a,b; initialized forecast skill = 0.88; detrended, initialized forecast skill = 0.66). Our comparison indicates that CESM-DPLE (and the reconstruction, for that matter) struggles to produce the pronounced trends toward anomalous $CO_2$ outgassing in the 1990s and anomalous $CO_2$ uptake in the 2000s. The ability (or lack thereof) of ESMs to reproduce the observationally-derived multi-decadal air-sea $CO_2$ flux variability has been the subject of recent publications (e.g., Li and Ilyina, 2018; Gruber et al., 2017), though no robust mechanisms seem to explain the (mis)match. The CESM-DPLE initialized forecast in forecast lead year 1 exhibits moderate predictive skill in the tropics and subtropics (Figure 7), and low skill elsewhere.

## 3.3 Predictability and predictive skill on the biome scale

Because the predictability of air-sea $CO_2$ flux is primarily driven by predictability of the biogeochemical state variables DIC and Alk, it makes sense to aggregate predictability across biogeographical biomes. We probe the limits of predictability and predictive skill in regional air-sea $CO_2$ flux by averaging the local flux across 17 biogeographical biomes. This is achieved by re-gridding the Fay and McKinley (2014) mean biome mask to the CESM model grid and computing the area-weighted, average $CO_2$ flux from the reconstruction, CESM-DPLE initialized forecasts, and observationally-derived $pCO_2$ product. The detrended $CO_2$ flux anomalies for three of the biomes are shown for forecast lead year 1 in Figure 8, and the predictability and predictive skill across all biomes is detailed in Table 1. These three biomes were chosen to contrast their predictability and/or predictive skill.

The biome-averaged $CO_2$ flux anomalies from the CESM-DPLE initialized forecast in forecast lead year 1 exhibit high correlations with the reconstruction anomalies in the North Pacific Subtropical biomes, and in the Southern Ocean Ice biome (Figure 8; Table 1), indicating high potential for prediction of $CO_2$ flux anomalies. This predictability decreases with increasing forecast lead time in the North Pacific Subtropical biomes, but persists for the Southern Ocean ice biome through forecast years 7-9 (Figure 8). Indeed, the Southern Ocean Ice biome is an anomaly in this regard; in the other 16 biomes, predictability drops off with prediction lead time (not shown).

Initialization engenders predictability of air-sea $CO_2$ flux variability the North Pacific Subtropical biomes, as we find low correlation between the uninitialized CESM-LE forecast $CO_2$ flux anomalies and the reconstruction anomalies here (Figure 8a,b; Table 1). The initialized forecast for these biomes has higher predictability than the uninitialized forecast and the persistence forecast for 7-8 years (Figure 9). These conclusions hold for most of the other ocean biomes (Table 1), with a few exceptions where the uninitialized forecast and/or persistence forecast are similar to the initialized forecast (e.g., the East Pacific Equatorial biome). In the Southern Ocean Ice biome, the $CO_2$ flux predictability is almost entirely driven by external forcing, and the persistence forecast indicates high predictability, as well (Figure 8; Figure 9, Table 1). Thus, the high and long-lasting predictability in this biome must be interpreted with caution, given the importance of external forcing in predicting $CO_2$ flux anomalies here.

The predictive skill of CESM-DPLE in forecast lead year 1 is illustrated for three biomes in Figure 8 and Table 1. Again, we note the moderate skill in the tropics and subtropics, and lower skill elsewhere.



The difference in the predictability between the initialized forecasts and the uninitialized and persistence forecasts reveals the impact of initialization on predictions of air-sea $CO_2$ flux variability on the biome scale (Figure 9). We probe the limits of initialized predictability in each biome by calculating the maximum forecast lead time for which the initialized CESM-DPLE $CO_2$ flux forecast has both higher predictability than the uninitialized CESM-LE and persistence forecasts and present the results in Figure 10. Our results indicate that initialization improves the forecast for the longest lead times in the subantarctic Southern Ocean and the northern North Atlantic, where the initialized forecast beats the other two forecasts out to forecast lead times of 10 and 9 years, respectively (Figure 10). Given the important role of these two regions for the global ocean uptake of anthropogenic carbon, and the numerous studies linking climate variability to air-sea $CO_2$ flux variability in these regions, this long-lasting predictability is akin to that of the global $CO_2$ flux integral and very encouraging. In other regions, however, such as the Southern Ocean Ice or East Equatorial Pacific biomes, the initialized forecast only beats the uninitialized or persistence forecast for a single year, indicating little benefit of forecast initialization on $CO_2$ flux forecasts here.

## 4   Conclusions

We analyze output from the CESM-DPLE system to quantify and understand the sources of predictability and predictive skill in global and regional air-sea $CO_2$ flux on annual to decadal timescales. We find high predictability in globally-integrated $CO_2$ flux that is engendered by initialization and extends to forecast lead times of $\sim$7 years. This predictability is evident across much of the global ocean, driven by predictability in $\Delta p CO_2$, which itself is primarily driven by predictability in surface ocean DIC and Alk. While the CESM-DPLE system exhibits strong predictability, model skill remains a challenge to developing useful forecasts. On the biome scale, we find particularly long-lasting predictability in the northern North Atlantic and subantarctic Southern Ocean that is engendered by initialization.

Our study complements two recent studies of ocean carbon decadal predictions conducted at different modeling centers. Li et al. (2016) use decadal predictions from MPI-ESM to investigate near-term changes in North Atlantic $CO_2$ flux, while Séférian et al. (2018) use CNRM-ESM1 to assess the predictability horizon of globally-integrated ocean and land carbon fluxes. While these studies use different prediction systems, we nevertheless come to some of the same conclusions. For example, Séférian et al. (2018) find that global ocean carbon uptake is potentially predictable for up to 6 years, and Li and Ilyina (2018) find high potential predictability in the North Atlantic that is engendered by initialization. These studies collectively suggest predictability for near-term ocean carbon uptake on global and regional scales, which is beneficial for forecasting the future global carbon budget and climate system.

*Acknowledgements.* The CESM-DPLE was generated using computational resources provided by the National Energy Research Scientific Computing Center, which is supported by the Office of Science of the U.S. Department of Energy under Contract No. DE-AC02-05CH11231, as well as by an Accelerated Scientific Discovery grant for Cheyenne (doi:10.5065/ D6RX99HX) that was awarded by NCAR's Computational and Information Systems Laboratory. The NCAR contribution to this study was supported by the National Oceanic and Atmospheric Administration Climate Program Office under Climate Variability and Predictability Program grant NA09OAR4310163, the National Sci-



ence Foundation (NSF) Collaborative Research EaSM2 grant OCE-1243015, and the NSF through its sponsorship of NCAR. NSL is grateful for funding from NSF (OCE-1752724, OCE-1558225).



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


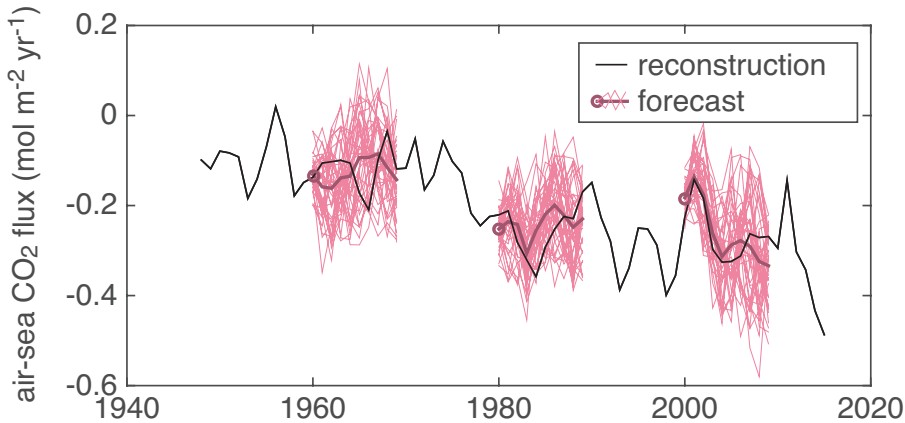

**Figure 1.** Annual mean air-sea $CO_2$ flux (mol m$^{-2}$ yr$^{-1}$) in the South Pacific subtropical permanently stratified biome for the (black) model reconstruction, and (pink) CESM-DPLE decadal forecasts initiated in 1960, 1980, and 2000 (other forecasts omitted for visual clarity). Thick magenta line represents the ensemble-mean forecast; open circles show the ensemble mean in forecast year 1. Positive fluxes denote ocean outgassing. Forecasts have been drift-corrected and adjusted to match the reconstruction climatological mean for ease of visual comparison.



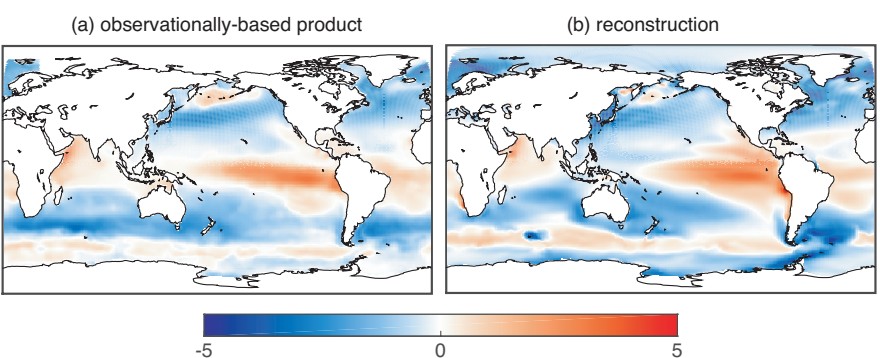

**Figure 2.** Annual-mean air-sea $CO_2$ flux (mol m$^{-2}$ yr$^{-1}$) over the period 1982-2015 as estimated by (a) the Landschützer et al. (2016) observationally-based product, and (b) the model reconstruction. Positive fluxes denote ocean outgassing.





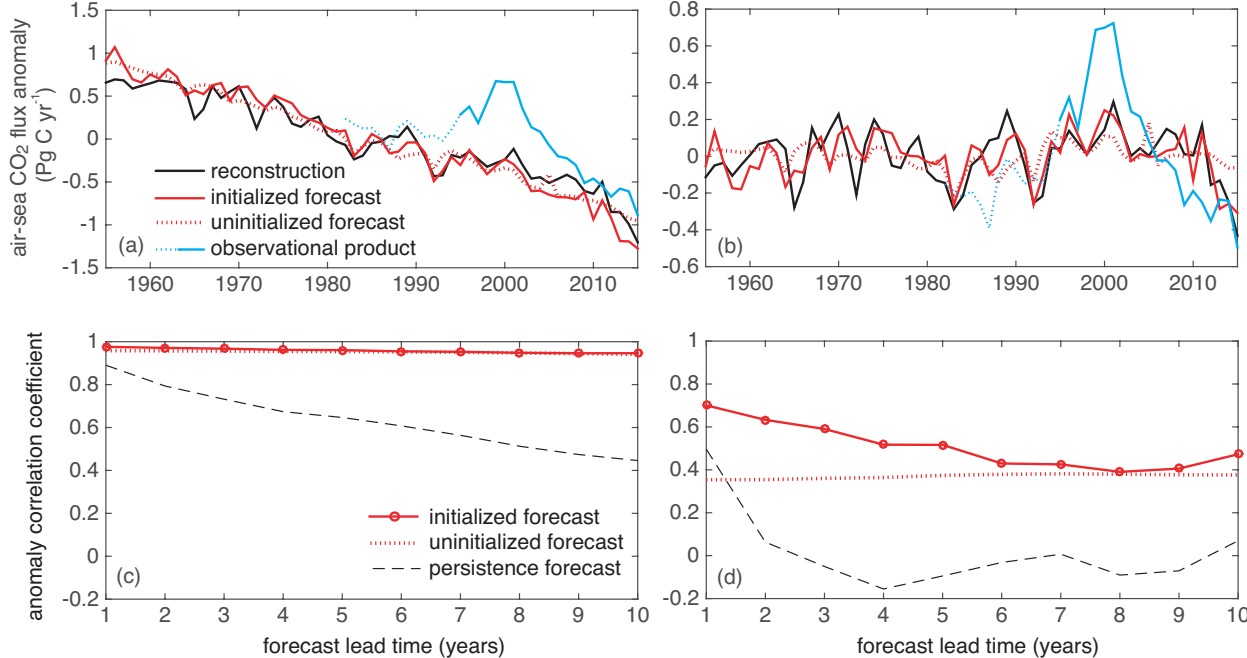

**Figure 3.** (a) Temporal evolution of the globally-integrated air-sea $CO_2$ flux anomaly, as estimated by the (black) reconstruction, (red) CESM-DPLE initialized forecast, (red dotted) CESM-LE uninitialized forecast, and (blue) Landschützer et al. (2016) observationally-based product. The CESM-DPLE time series is the drift-corrected, ensemble mean forecast anomalies over lead year 1, and the reconstruction, uninitialized forecast, and observational product have been transformed to anomalies by subtracting their respective climatological means. Observations prior to 1995 are dotted, due to lower observation density. Positive anomalies indicate anomalous ocean outgassing. (b) Same as (a), but with long-term linear trends removed from each time series. (c) Predictability of globally integrated $CO_2$ flux as a function of lead time, as indicated by the correlation coefficient of $CO_2$ flux anomalies from the (red) CESM-DPLE initialized forecast, and (red dotted) CESM-LE uninitialized forecast with the reconstruction. Black dashed line shows indicates the correlation coefficient of the persistence forecast as a function of lead time. Red circles on the initialized forecast indicate statistically significant predictability. (d) Same as (c), but with linear trends removed from each time series.

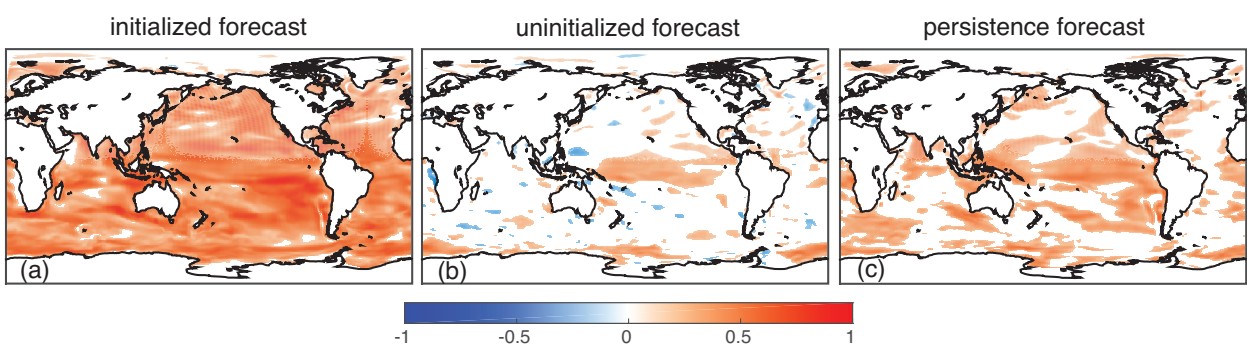

**Figure 4.** Predictability of air-sea $CO_2$ flux, as indicated by the correlation coefficient of detrended, air-sea $CO_2$ flux anomalies from the (a) CESM-DPLE initialized forecast lead year 1, and (b) CESM-LE uninitialized forecast with the reconstruction. (c) Correlation coefficient of the persistence forecast for lead year 1. Correlation coefficients that are not statistically significant are assigned a value of zero.





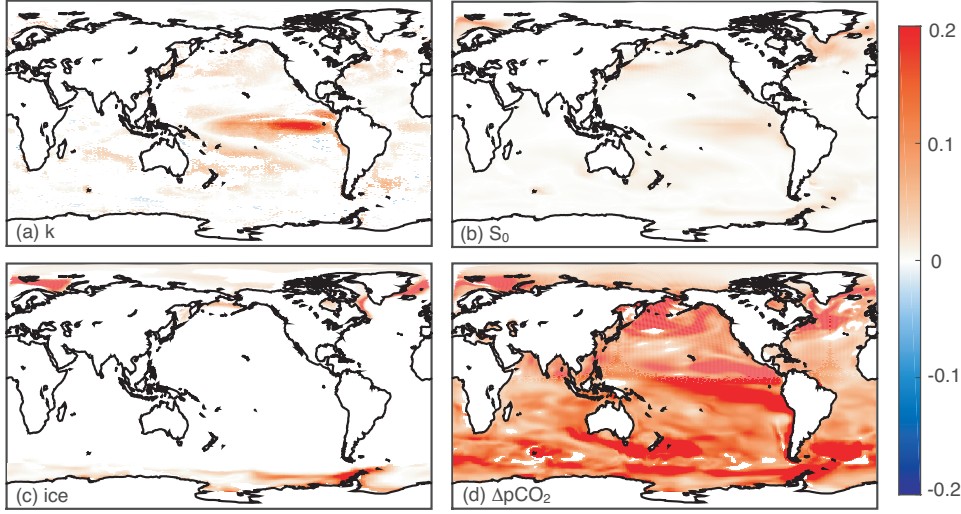

**Figure 5.** Drivers of predictability in air-sea $CO_2$ flux during forecast year 1, as indicated by the predictability of the (a) gas-exchange coefficient, (b) solubility, (c) sea ice fraction, and (d) $\Delta pCO_2$, scaled to $CO_2$ flux units (mol m$^{-2}$ yr$^{-1}$). Correlation coefficients that are not statistically significant are assigned a value of zero.




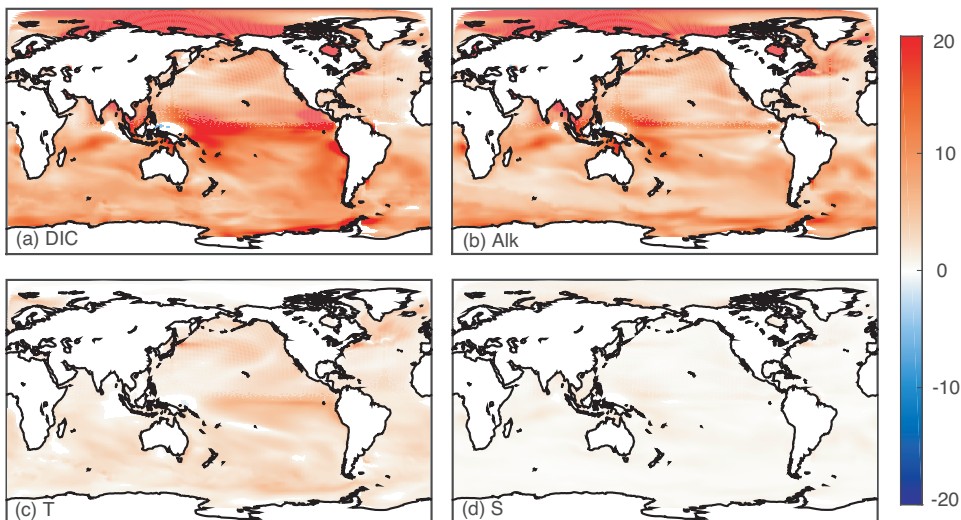

**Figure 6.** Drivers of predictability in surface ocean pCO₂ during forecast year 1, as indicated by the predictability of surface ocean (a) DIC, (b) Alk, (c) temperature, and (d) salinity, scaled to pCO₂ units ($\mu$atm). Correlation coefficients that are not statistically significant are assigned a value of zero.



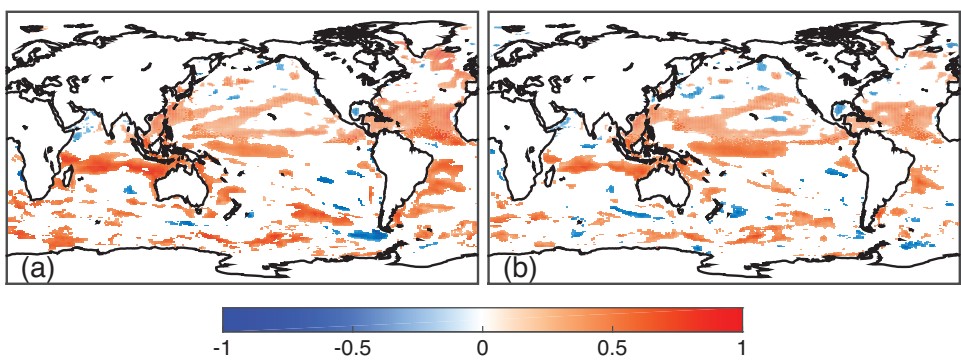

**Figure 7.** Air-sea $CO_2$ flux predictive skill, as indicated by the correlation coefficient of air-sea $CO_2$ flux (a) anomalies, and (b) linearly detrended anomalies from the CESM-DPLE initialized forecast in year 1 with the Landschützer et al. (2016) observational product over 1995-2015. Correlation coefficients that are not statistically significant are assigned a value of zero.



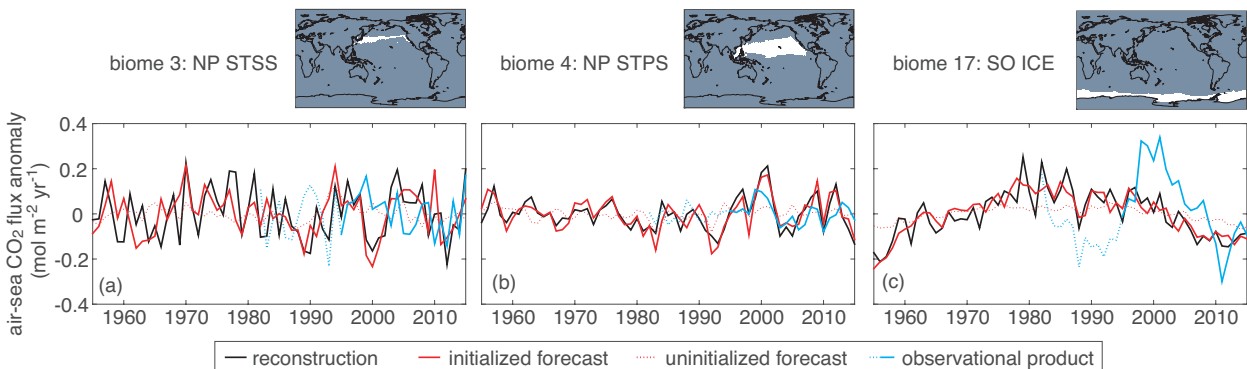

**Figure 8.** Temporal evolution of the biome-averaged air-sea $CO_2$ flux anomalies in the (a) NP STSS, (b) NP STPS, and (c) SO ICE biomes (mol m$^{-2}$ yr$^{-1}$). The following time series are plotted: (black) reconstruction, (red) CESM-DPLE initialized forecast, (red dotted) CESM-LE uninitialized forecast, and (blue) Landschützer et al. (2016) observationally-based product. The CESM-DPLE time series is the linearly detrended, drift-corrected, ensemble mean forecast anomalies in year 1; the reconstruction, CESM-LE ensemble mean, and observed time-series have been transformed to anomalies by removing the linear trend. Observations prior to 1995 are dotted, due to lower observation density. Positive anomalies indicate anomalous ocean outgassing.





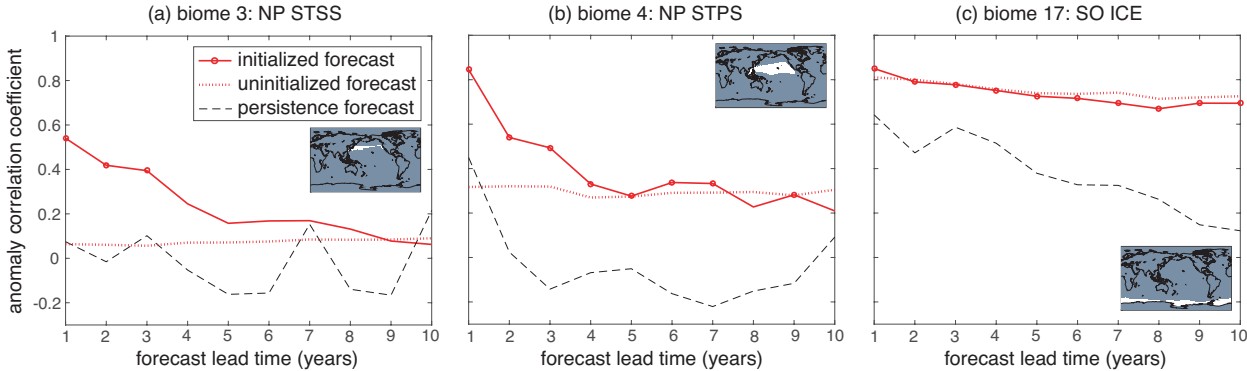

**Figure 9.** Predictability of biome-average $CO_2$ flux as a function of lead time in the (a) NP STSS, (b) NP STPS, and (c) SO ICE biomes, as indicated by the correlation coefficient of $CO_2$ flux anomalies from the (red) CESM-DPLE initialized forecast, and (red dotted) CESM-LE uninitialized forecast with the reconstruction. Black dashed line shows the correlation coefficient of the persistence forecast as a function of lead time. Red circles on the initialized forecast indicate statistically significant predictability.



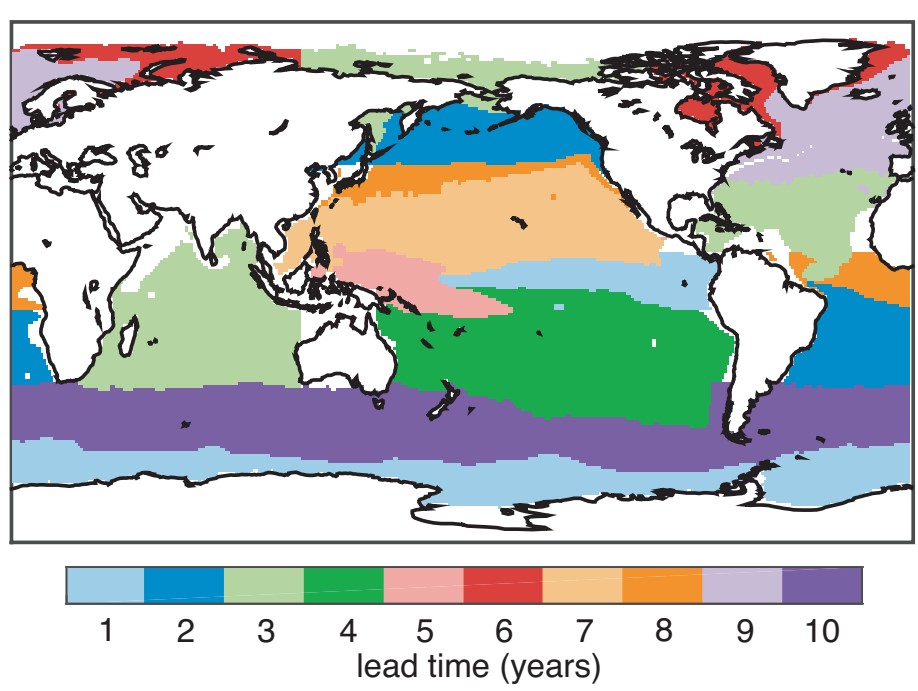

**Figure 10.** For each biome, the maximum forecast lead time (years) in which the initialized CESM-DPLE $CO_2$ flux forecast has both higher predictability than the uninitialized CESM-LE forecast and a higher correlation coefficient than the persistence forecast.





**Table 1.** Predictability and predictive skill of biome-averaged air-sea $CO_2$ flux, as indicated by correlation coefficients of flux anomalies for forecast year 1.

| Biome Name | Biome Acronym | Biome Number | Initialized Forecast | Uninitialized Forecast | Persistence Forecast | Forecast Skill |
|---|---|---|---|---|---|---|
| North Pacific Ice | NP ICE | 1 | 0.29 | -0.22 | 0.25 | 0.43 |
| North Pacific Subpolar Seasonally Stratified | NP SPSS | 2 | 0.54 | -0.12 | 0.47 | -0.45 |
| North Pacific Subtropical Seasonally Stratified | NP STSS | 3 | 0.54 | 0.06 | 0.07 | -0.28 |
| North Pacific Subtropical Permanently Stratified | NP STPS | 4 | 0.85 | 0.32 | 0.45 | 0.60 |
| West Pacific Equatorial | PEQU-W | 5 | 0.73 | 0.31 | 0.52 | 0.66 |
| East Pacific Equatorial | PEQU-E | 6 | 0.64 | 0.35 | 0.50 | 0.53 |
| South Pacific Subtropical Permanently Stratified | SP STPS | 7 | 0.81 | 0.33 | 0.50 | 0.19 |
| North Atlantic Ice | NA ICE | 8 | 0.49 | 0.07 | 0.24 | 0.36 |
| North Atlantic Subpolar Seasonally Stratified | NA SPSS | 9 | 0.55 | 0.10 | 0.17 | -0.28 |
| North Atlantic Subtropical Seasonally Stratified | NA STSS | 10 | 0.53 | -0.08 | 0.01 | -0.10 |
| North Atlantic Subtropical Permanently Stratified | NA STPS | 11 | 0.72 | 0.35 | 0.18 | 0.56 |
| Atlantic Equatorial | AEQU | 12 | 0.55 | 0.17 | 0.27 | -0.04 |
| South Atlantic Subtropical Permanently Stratified | SA STPS | 13 | 0.60 | 0.09 | 0.16 | 0.49 |
| Indian Ocean Subtropical Permanently Stratified | IND STPS | 14 | 0.16 | -0.11 | 0.05 | 0.31 |
| Southern Ocean Subtropical Seasonally Stratified | SO STSS | 15 | 0.70 | -0.02 | 0.20 | 0.26 |
| Southern Ocean Subpolar Seasonally Stratified | SO SPSS | 16 | 0.47 | 0.08 | 0.32 | 0.47 |
| Southern Ocean Ice | SO ICE | 17 | 0.85 | 0.81 | 0.64 | 0.60 |