# Peer review of "Predicting near-term variability in ocean carbon uptake"

_Earth System Dynamics, 2018_

## Referee Comment (RC1) · Anonymous Referee #1 · 30 Oct 2018

In the manuscript 'Predicting near-term changes in ocean carbon uptake' Lovenduski and coauthors assess the predictability of the ocean carbon sink over the last decades using CESM-DPLE, a new large ensemble decadal prediction platform developed at NCAR. By realizing 40 decade-long ensemble members each year from 1954 to 2015, the authors estimate that the global ocean carbon sink is predictable up to 7 years in advance which is in the line of recent published estimates. The authors also investigate the drivers of this predictability and explain that it arises from the predictability of carbon-related fields (DIC and Alkalinity, setting $\Delta pCO2$ and hence carbon fluxes).

The paper is well-written and the analyses are sounds. I much appreciate this work which explores the predictive capability of the current generation of ESM. Nevertheless, I think this paper needs some clarification that have to be addressed first, and which

prevent me of accepting this paper in its present form.

General Major Comments: 1- My first comment concerns the assessment of the initialization procedure which is the core of decadal prediction. Here the other briefly describe this step but do not provide a complete evaluation. ESD paper are not limited by the length. Thus I recommend to include a new section to discuss the initialization procedure because: (1) full fields restoring implies a model drift; this is an interesting to document how it impacts the biogeochemical fields, especially the carbon related fields and nutrients fields that could generates non-linearity in the drift (2) your initialization strategy fails at capturing the recent variability in the ocean carbon sink as suggested by SOM-FFN dataproduct. It could be interesting to show other variables such as SST or the AMOC to support the fact that your initialization procedure is doing a good job. 2- Further discussions is needed when discussing the drivers. It could be interesting to document if your model gives a longer predictability horizon for SST, DIC, Alk . . . than that of ocean carbon sink and compares this result to the persistence. At least for DIC and Alk which have a long-lasting memory it would be helpful to demonstrate that your model's predictability beats the persistence for those fields otherwise it might suggest that the predictability in ocean carbon sink is supported by the persistence of DIC and Alkalinity anomalies. 3- Finally, further discussions are needed to discuss how this work compared with previous works based on different prediction system such as Li et al. (2016), Séférian et al., (2018) [using ESMs] but also all the recent studies focusing on ocean physical variables (e.g., Kim et al., 2012)

Specific comments: Page 1 Title: I suggest to modify the text because "near-term changes" could also implies anthropogenic carbon sink. This latter is rather well captured by ESM without initialization. I suggest to use "multi-annual variations" instead. L2 : please define somewhere what your mean by "near-term" L4: of an Earth system model L4: initialized forecast=please explain the initialization somewhere in the abstract and avoid the terminology initialized forecast because of forecast is generally initialized L9: moderate predictive skill= please explain the predictive skill measure L11:

[Figure]

initialized predictability= predictability also implies initialization. I suggest to remove 'initialized' L21: I suggest to include observational references instead (e;g., Landshutzer et al., 2016)

Page 2 L7-15 Please expand the discussion by including the key limits of the decadal predictability that were highlight in the literature. For example, the first attempt from Keenlyside et al and Smith et al in 2008 which were challenged 10 years after by the recent observations. Besides, you could include a better rationale of the first attempt in the Earth system community such as Li et al. 2016, Séférian et al. 2018 for the ocean carbon sink and Séférian et al. 2014 for the net marine productivity. And the use of statistical model such as Betts et al. (2016, 2018) for atmospheric CO2. L15 Please add Resplandy et al. (2014) which describes how far the decadal variability in ocean carbon fluxes differs between models

Page 3 L7-13: Please expand this paragraph— see my major comments L20: a reasonable job is not enough to determine if a model is fitted for purpose. Could you please provide further details such as the spatial correlation, the RMSE . . . L23: 10-14 Kelvin ? This is a really small perturbation. Have assessed if this initial perturbation lead to populate the full range of model variability as diagnosed from the piControl ? It is important to tell the reader if you chose to populate this uncertainty or instead to stay close from the initial conditions L33: Please provide a figure of the ensemble with the drift as in Kim et al 2012 in addition to the de-drifted ensemble. You could add on panel on figure 1 to show that.

Page 4 L9-11: I'm a bit puzzled here. Unless I misunderstood McKinley et al 2016 and Lovenduski et al. 2016 used a CMIP5-style CESM and hence CMIP5 forcings to performed their analyses. Here, CESM-DPLE is setup for CMIP6 and hence use CMIP6 forcings, right ? If it does, several external forcing have been revised between CMIP5 and CMIP6. This is the case for the volcanoes (which influence the predictability). Could you please comment this point ?

Page 5 L10-11 What happens if you consider the full observational time-series ? Besides could you please explain why the correlation of the uninitialized simulation slightly increases with lead time ? On Figure 3 8 and 9 please indicate the correlation limits (R*) on the graphs and indicates the level of confidence (and the number of degree of freedom used for the t-test) employed; this information is missing. L15: why are you talking about emissions and terrestrial $CO_2$ uptake. Your model set-up employs $CO_2$ concentration as prescribed in the forcing, correct ? L25: linearly detrended forecast: have you done this detrended at grill-cell scale or have you applied the same detrending globally ? L26: as you suggested that your modelling plateform is able to predict ocean carbon sink up to 7 years in advance it could be usefull to show what happens at lead time greater than 1 year. Could you please replace the figures showing lead-time (LT) 1 by LT 7 and/or moving LT1 Figures as supplemental data

Page 7 L2: please explain what are your forecast skill and what is the limit for a skillfull predictability at a given confidence L11: predictability or persistence — please see my major comments L23: You could state that this biomes does not see an impacts of your initialization procedure. Maybe the sea-ice influence regions are not restored to the observations? This is why I suggest to further develop this point with a new section in the ms— see my major comments

Page 8 L1: forecast and the uninitialized= remove 'forecast and' L15-21: Please further discuss the limit of your approach= for example your initialization procedure fails at capturing the observed variability. You estimate a predictability of 7 years with only 20 years of data (which is not enough). I suggest the authors to discuss this point and hence to highlight the most of the results presented in this work relates to the potential predictability rather than an effective predictability.

Betts, R. A., Jones, C. D., Knight, J. R., Keeling, R. F., Kennedy, J. J., Wiltshire, A. J., . . . Aragão, L. E. O. C. (2018). A successful prediction of the record CO 2 rise associated with the 2015/2016 El Niño. Philosophical Transactions of the Royal Society B: Biological Sciences, 373(1760), 20170301. https://doi.org/10.1098/rstb.2017.0301

[Figure]

Betts, R. A., Jones, C. D., Knight, J. R., Keeling, R. F., & Kennedy, J. J. (2016). El Niño and a record CO2 rise. Nature Climate Change, 6(9), 806–810. https://doi.org/10.1038/nclimate3063 Keenlyside, N. S., Latif, M., Jungclaus, J., Kornblueh, L., & Roeckner, E. (2008). Advancing decadal-scale climate prediction in the North Atlantic sector. Nature, 453(7191), 84–88. https://doi.org/10.1038/nature06921 Kim, H. M., Webster, P. J., & Curry, J. A. (2012). Evaluation of short-term climate change prediction in multi-model CMIP5 decadal hindcasts. Geophysical Research Letters, 39(10), L10701. https://doi.org/10.1029/2012GL051644 Li, H., Ilyina, T., Müller, W. A., & Sienz, F. (2016). Decadal predictions of the North Atlantic CO2 uptake. Nature Communications, 7(May 2015), 11076. https://doi.org/10.1038/ncomms11076 Resplandy, L., Séférian, R., & Bopp, L. (2015). Natural variability of $CO_2$ and $O_2$ fluxes: What can we learn from centuries-long climate models simulations? Journal of Geophysical Research: Oceans, 120(1). https://doi.org/10.1002/2014JC010463 Séférian, R., Berthet, S., & Chevallier, M. (2018). Assessing the Decadal Predictability of Land and Ocean Carbon Uptake. Geophysical Research Letters, 45(5), 2455–2466. https://doi.org/10.1002/2017GL076092 Séférian, R., Bopp, L., Gehlen, M., Swingedouw, D., Mignot, J., Guilyardi, E., & Servonnat, J. (2014). Multiyear predictability of tropical marine productivity. Proceedings of the National Academy of Sciences of the United States of America, 111(32). https://doi.org/10.1073/pnas.1315855111 Smith, D. M., Cusack, S., Colman, A. W., Folland, C. K., Harris, G. R., & Murphy, J. M. (2007). Improved Surface Temperature Prediction for the Coming Decade from a Global Climate Model. Science, 317(5839), 796–799. https://doi.org/10.1126/science.1139540

---

## Referee Comment (RC2) · Anonymous Referee #2 · 5 Nov 2018

The authors have investigated the predictability and predictive skill of the ocean carbon uptake by using a large ensemble of 40-member decadal prediction and historical simulations based on NCAR CESM. They found a prominent improved predictability of the ocean carbon uptake in the initialized simulations in comparing with the uninitialized historical simulations and the persistence forecast. Furthermore, they attribute the predictability of ocean carbon uptake to the dissolved inorganic carbon and alkalinity. The outcome of this study is an important contribution for understanding and predicting variations of the ocean carbon uptake and the global carbon cycle, which are crucial for estimating climate change. Moreover, reconstruction and near-term predictions of global carbon cycle show large potential for supporting the future carbon stocktaking. Therefore the study on this topic merits publication on the Earth System Dynamics.

[Figure]

The manuscript is well written and the results are clearly stated, however, the conclusions are not quantitatively precise and not statistically robust from the results. This together with some other issues listed below prevents me accepting this manuscript at its present format.

1. The authors claimed a potential predictive skill of up to 7 years in the abstract and conclusions. Is this conclusion from Fig. 3d by comparing the initialized forecast to the uninitialized forecast? The authors did not do a statistical test if the difference between initialized and the uninitialized forecast is significant. The red circles only show if the correlation of the initialized forecast itself is significant. It seems to me that the initialized forecast (red dots) at lead time of 6 and 7 years are very close to the uninitialized forecast, these are probably not significantly distinguishable. As the improved skill due to initialization is a main quantitative conclusion in this manuscript, it requires a sound significant test, such as the commonly used bootstrap method (Goddard et al., 2013), which is also suggested by the Decadal Climate Prediction Project (DCPP) (Boer et al., 2016). In addition, the authors only show time-series and maps of predictability at lead time of 1 year. Given the high predictability of ocean carbon uptake as stated in this study, time-series and maps of predictability at longer lead time at least of 2 years are more representative.

2. The authors estimated both potential predictability against reconstruction and predictive skill against observation-based data product. The two results were separately discussed in the main text, however, the conclusions are mixed especially in the abstract. It's quite difficult for the readers to distinguish the origin of the conclusions, they are from potential skill or skill against observation. For this reason, the abstract needs to be reorganized and make it clearer. Furthermore, the connections between potential predictabiltiy and predictive skill are weak in the manuscript. How consistent/inconsistent are the predictability and the predictive skill? What would be the implication of potential predictability to the predictive skill versus observation?

3. The initialized simulations were started from a forced ocean-sea ice simulation for

the ocean-sea ice component, but were started from the CESM Large Ensemble for the atmosphere and the land components (details were described in page 3 lines 8-13). This means that the ocean and the atmosphere and land are most probably in different climate state/phase, they need to adjust to each other and approach a new equalibrium. The mismatch of initial conditions in the ocean and in the atmosphere and land would affect the variations and predictions of the system, especially for the carbon flux across the boundaries. Discussions of the effects of mismatch in the ocean and the atmophere and land are necessary. Can the model drift due to the mismatch be largely eliminated by the drift correction?

4. As stated in McKinley et al. (2016), some ensemble members of the CESM-LE have problem in the ocean biogeochemical outputs. McKinley et al. (2016) used only 32 ensemble members of the CESM-LE, because some ensemble members were discarded due to a setup error which leads to corrupts of ocean biogeochemical output. In this study, the authors use 40 ensemble members as written in Page 4 lines 5-9. How do the authors treat the ensemble members with setup error in this study?

5. The numbers in Fig. 10 are not significant and deducible from Fig. 9. For instance, the maximum forecast lead time in biome 3 (NP STSS) is 8 years in Fig. 10, but if we look at Fig. 9a, the correlations at lead time beyond 4 years are not significant and end up with less than 0.2 at lead time of 8 years. As for biome 4 (NP STPS), the maximum forecast lead time is 7 years in Fig. 10, but the initialized forecast skill is not significantly higher than the uninitialized forecast skill at lead time of 5 years in Fig. 9b. Therefore, I think the numbers in Fig. 10 need to be carefully checked by taking into account the significant test and the relative magnitude of the correlations.

6. Table 1: the table caption and the title of the columns are unclear. I guess the "Initialized forecast" and the "uninitialized forecast" refer to forecast skill versus reconstruction, and the "Forecast skill" refer to forecast skill versus observation-based products. The time period used to calculate the correlations needs to be specified, especially for the "Forecast skill" which use much shorter period. In addition, statistical significant

test information by highlighting of the numbers will be also helpful. Moreover, a table of predictability for the maximum forecast lead time will be necessary as supplementary information to Fig. 10.

7. Fig. 2: how different is the reconstruction comparing to the uninitialized simulations? Is the reconstruction closer to observations than the uninitialized simulaitons? It would be more informative to also include the climatology of the uninitialized simulations.

8. It is not introduced but I guess the authors use different time period for the drift correction and correlation calculation along different lead time. As shown in Fig. 3d, the red dashed line has a slightly positive trend, which indicates that the authors use different time period for the correlation calculation for different lead time. To make a consistent estimate of predictive skill along all the forecast range, it is better to use the same time period for all the lead years as suggested by DCPP (Boer et al., 2016, Appendix E) and previous studies focusing on the physical predictions (Hawkins et al., 2014; Smith et al., 2013).

9. Page 3 line 7: are the historical external forcings from CMIP5 or CMIP6 (i.e., the 5th of 6th Coupled Model Intercomparison Project)?

10. Page 5 line 16: "...for those forecasting year-to-year changes..." should be "...for those reproducing year-to-year changes..."

11. Page 6 line 5: "...the anomaly correlation coefficients are scaled to CO2 flux units..." The correlation coefficient itself is uniformed and has no unit, there is no need to further scale it. What are the results based on the correlation coefficients without the scaling? I think the results without scaling are similar to those based on the scaled correlations. It worths to check. One more question on the scaling formular: how do the authors calculate the $\partial\Phi/\partial x$, how long is the time step?

12. Page 6 line 22-23: "The similar predictability of DIC and Alk across many regions hints at an important role for ocean circulation, rather than biological productivity...,

in CO2 flux predictability." From this I understand that the biological productivity is a secondary regulation of CO2 flux, therefore the biome division is probably not a proper way to divide the global ocean for CO2 flux predictions. The last sentence is the same as line 8-9 on Page 7.

13. Page 8 line 26: "Li and Ilyina (2018)" should be "Li et al. (2016)", right?

14. Figure 4 caption: "CESM-DPLE initialized forecast lead year 1" needs to be revised and includes information of the counterpart of the correlation, e.g., "CESM-DPLE initialized forecast for lead year 1 with the reconstruction".

15. Figure 9: are the correlations based on detrended time series?

References: Boer, G. J., et al.: The Decadal Climate Prediction Project (DCPP) contribution to CMIP6, Geosci. Model Dev., 9, 3751-3777, https://doi.org/10.5194/gmd-9-3751-2016, 2016. Goddard, L., et al.: A verification framework for interannual to decadal predictions experiments, Clim. Dynam., 40, 245–272, doi:10.1007/s00382-012-1481-2, 2013. Hawkins, E., Dong, B., Robson, J., Sutton, R., and Smith, D.: The interpretation and use of biases in decadal climate predictions, J. Climate, 27, 2931–2947, doi:10.1175/JCLI-D-13-00473.1, 2014. McKinley, G. A., Pilcher, D. J., Fay, A. R., Lindsay, K., Long, M. C., and Lovenduski, N. S.: Timescales for detection of trends in the ocean carbon sink, Nature, 530, 469–472, http://dx.doi.org/10.1038/nature16958, 2016. Smith, D. M., Eade, R., and Pohlmann, H.: A comparison of fullfield and anomaly initialization for seasonal to decadal climate prediction, Clim. Dynam., 41, 3325–3338, doi:10.1007/s00382-013-1683-2, 2013.
* * *

---

## Author Comment (AC1) · 14 Dec 2018

The comment was uploaded in the form of a supplement:
https://www.earth-syst-dynam-discuss.net/esd-2018-73/esd-2018-73-AC1-supplement.pdf

––––––––––––––––––––––––––––––

---

## Author Comment (AC2) · 14 Dec 2018

Response to Referees

We are extremely grateful to the two anonymous reviewers for providing such thorough reviews of our manuscript and suggesting changes that have substantially improved the science. Below, we include the original reviewer comments in black, with our response below in red.

We have made the following substantial changes to the manuscript:

1) We now more thoroughly assess the statistically significance of predictability from the initialized forecasts relative to that of the uninitialized and persistence forecasts using a z-test statistic for the 95% confidence interval. All figures and tables now report the results of this significance test.

2) We include a supplemental figure that illustrates air-sea $CO_2$ flux predictability for lead years 2 through 10, as requested by both reviewers.

3) We have clarified the difference between the potential predictability and the skill relative to observational metrics.

4) We now adhere to the DCPP protocols for calculating the predictability from the uninitialized correlation, as suggested by both reviewers.

**Anonymous Referee #1**

In the manuscript 'Predicting near-term changes in ocean carbon uptake' Lovenduski and coauthors assess the predictability of the ocean carbon sink over the last decades using CESM-DPLE, a new large ensemble decadal prediction platform developed at NCAR. By realizing 40 decade-long ensemble members each year from 1954 to 2015, the authors estimate that the global ocean carbon sink is predictable up to 7 years in advance which is in the line of recent published estimates. The authors also investigate the drivers of this predictability and explain that it arises from the predictability of carbon-related fields (DIC and Alkalinity, setting _pCO2 and hence carbon fluxes).

The paper is well-written and the analyses are sounds. I much appreciate this work which explores the predictive capability of the current generation of ESM. Nevertheless, I think this paper needs some clarification that have to be addressed first, and which prevent me of accepting this paper in its present form.

General Major Comments:
1- My first comment concerns the assessment of the initialization procedure which is the core of decadal prediction. Here the other briefly describe this step but do not provide a complete evaluation. ESD paper are not limited by the length. Thus I recommend to include a new section to discuss the initialization procedure because: (1) full fields restoring implies a model drift; this is an interesting to document how it impacts the biogeochemical fields, especially the carbon related fields and nutrients fields that could generates non-linearity in the drift (2) your initialization strategy fails at capturing the recent variability in the ocean carbon sink as suggested by SOM-FFN dataproduct. It could be interesting to show other variables such as SST or the AMOC to support the fact that your initialization procedure is doing a good job.

We have added a more complete description/evaluation of the initialization procedure to Section 2 (see also our response to your Page 3, line 23 comment):

CESM-DPLE initializes an ensemble of 40 simulations each year using round-off level (order $10^{-14}$) perturbations in the initial air temperature field (Figure 1). Previous work indicates that this small perturbation in the initial conditions generates a wide divergence in global mean surface temperatures across the ensemble members within about 30 days (V. Yettella, pers. comm., 2018), and the average divergence in globally-integrated, annual-mean forecast $CO_2$ flux across the ensemble members (0.53 Pg C $yr^{-1}$) is an order of magnitude greater than that generated by the preindustrial control simulation of CESM (0.09 Pg C $yr^{-1}$; Lovenduski et al.,2015b).

Indeed, the full-field initialization procedure generates a drift in the forecasts. As stated in the original manuscript, we correct this drift by transforming to anomalies from a drifting climatology. The drifting climatology is statistically robust due to the large number (n=40) of ensemble members in each forecast and the large number (n=62) of start years. In the revised

manuscript, we now note that the drifting climatology need not be linear, as its temporal behavior is determined by the variable of interest:

*Note that this method does not assume that the drift is linear, and disregards potential dependence of the drift on the external forcing.*

We include for the reviewer the companion to Figure 1 that shows the non-drift corrected forecast (see Figure R1 below). It is nearly identical to the original Figure 1, indicating that the drift in $CO_2$ flux is not particularly large, and thus further description of the drift is unnecessary.

[Figure]

**Figure R1**. Annual-mean air-sea $CO_2$ flux (mol m$^{-2}$ yr$^{-1}$) in the South Pacific subtropical permanently stratified biome for the (black) model reconstruction and (pink) CESM-DPLE decadal forecasts initiated in 1960, 1980, and 2000 (other forecasts omitted for visual clarity). Thick magenta line represents the ensemble-mean forecast; open circles show the ensemble mean in forecast year 1. Positive fluxes denote ocean outgassing. Forecasts have not been drift corrected.

We note that the reconstruction fails to capture the recent variability in air-sea $CO_2$ flux (compare black and blue lines in Figures 3a and 3b), so the initialization procedure for the forecast is not to blame for the poor skill.

We include for the reviewer below Figure R2 from Yeager et al. (2018) which illustrates the forecast skill of the CESM-DPLE SST over a range of lead years. This figure demonstrates that the initialization improves the skill of the SST forecast.

[Figure]

**Figure R2**. (a) – (c) Anomaly correlation coefficient of annual SST from CESM-DPLE relative to ERSSTv5 observations for lead times of 1-5, 3-7, and 5-9 years, respectively. Anomaly correlation coefficient skill score differences (d) – (f) between CESM-DPLE and persistence and (g) – (i) between CESM-DPLE and CESM-LE. From Yeager et al. (2018).

2- Further discussions is needed when discussing the drivers. It could be interesting to document if your model gives a longer predictability horizon for SST, DIC, Alk : : : than that of ocean carbon sink and compares this result to the persistence. At least for DIC and Alk which have a long-lasting memory it would be helpful to demonstrate that your model's predictability beats the persistence for those fields otherwise it might suggest that the predictability in ocean carbon sink is supported by the persistence of DIC and Alkalinity anomalies.

Indeed this would be interesting -- one could imagine performing a separate analysis to determine the predictability horizons of all relevant physical and biogeochemical variables, but we feel this is beyond the scope of our study. Here, we have demonstrated that the air-sea $CO_2$ flux predictability originates from $pCO_2$ and is linked to predictability in DIC and Alk – thus, if one can accurately predict DIC and Alk, one can then predict $CO_2$ flux variability. Perhaps a future publication could explore this topic in greater detail (see response to point 3 below). Manuscript unchanged in response to comment.

3- Finally, further discussions are needed to discuss how this work compared with previous works based on different prediction system such as Li et al. (2016), Séférian et al., (2018) [using ESMs] but also all the recent studies focusing on ocean physical variables (e.g., Kim et al., 2012)

We agree that a discussion of the various prediction systems and their findings for $CO_2$ flux is important, and we report the conclusions of these first two studies in our Conclusions section. Two of us (Lovenduski and Yeager) are co-authors on a review paper being drafted on this very topic (lead author is Tatiana Ilyina, journal is Current Climate Change Reports) that will cover the differences between the decadal prediction systems at the various modeling centers and the main findings for ocean carbon uptake. Perhaps this review paper could also include an analysis of the predictability horizons (as suggested in the previous comment). Alas, we await the findings of this review. Manuscript unchanged in response to comment.

Specific comments:
Title: I suggest to modify the text because "near-term changes" could also implies anthropogenic carbon sink. This latter is rather well captured by ESM without initialization. I suggest to use "multi-annual variations" instead.

Thank you for the suggestion. We have modified the title to read "Predicting near-term variability in ocean carbon uptake".

L2 : please define somewhere what your mean by "near-term"

Good catch. We have modified the first sentence of the abstract to read:

*Interannual variations in air-sea fluxes of carbon dioxide ($CO_2$) impact the global carbon cycle and climate system, and previous studies suggest that these variations may be predictable in the near-term (from a year to a decade in advance).*

L4: of an Earth system model

Manuscript changed as suggested.

L4: initialized forecast=please explain the initialization somewhere in the abstract and avoid the terminology initialized forecast because of forecast is generally initialized

We have excised the word "initialized" from the abstract. The abstract now reads:

*Interannual variations in air-sea fluxes of carbon dioxide ($CO_2$) impact the global carbon cycle and climate system, and previous studies suggest that these variations may be predictable in the near-term (from a year to a decade in advance). Here, we quantify and understand the sources of near-term predictability and predictive skill in air -sea $CO_2$ flux on global and regional scales by analyzing output from a novel set of retrospective decadal forecasts of an Earth system model. These forecasts exhibit the potential to predict year-to-year variations in the globally-integrated air-sea $CO_2$ flux several years in advance, as indicated by the high correlation of the forecasts with a model reconstruction of past $CO_2$ flux evolution. This potential predictability exceeds that obtained solely from foreknowledge of variations in*

*external forcing or a simple persistence forecast, with the longest-lasting forecast enhancement in the subantarctic Southern Ocean and the northern North Atlantic. Potential predictability in $CO_2$ flux variations are largely driven by predictability in the surface ocean partial pressure of $CO_2$, which itself is a function of predictability in surface ocean dissolved inorganic carbon and alkalinity. The potential predictability, however, is not realized as predictive skill, as indicated by the moderate to low correlation of the forecasts with an observationally-based $CO_2$ flux product. Nevertheless, our results suggest that year-to-year variations in ocean carbon uptake have the potential to be predicted well in advance, and establish a precedent for forecasting air-sea $CO_2$ flux in the near future.*

L9: moderate predictive skill= please explain the predictive skill measure

We now indicate the measure (correlation) in the abstract -- see above.

L11: initialized predictability= predictability also implies initialization. I suggest to remove 'initialized'

We have excised the word "initialized" from the abstract – see above.

L21: I suggest to include observational references instead (e;g., Landshutzer et al., 2016)

We have added a reference to Landschutzer et al. (2016) here.

L7-15 Please expand the discussion by including the key limits of the decadal predictability that were highlight in the literature. For example, the first attempt from Keenlyside et al and Smith et al in 2008 which were challenged 10 years after by the recent observations. Besides, you could include a better rationale of the first attempt in the Earth system community such as Li et al. 2016, Séférian et al. 2018 for the ocean carbon sink and Séférian et al. 2014 for the net marine productivity. And the use of statistical model such as Betts et al. (2016, 2018) for atmospheric CO2.

We now make reference to these first attempts at decadal prediction (Smith et al., 2007; Keenlyside et al., 2008) in the introduction. In addition to referencing the Li and Seferian studies, we now also include a discussion of the limitations of decadal predictability in the Conclusions section:

*While the ever-expanding field of decadal climate prediction has the potential to inform policy and management decisions moving forward, decadal forecasts come with several caveats. Initialization shock and drift of the coupled model system, inability of Earth system models to realistically simulate internal variability, uncertain future levels of radiative forcing, and imperfect observations are frequently cited as limitations to making accurate forecasts of the future (Meehl et al., 2014). In the case of ocean carbon, it is important to note that potential predictability in regional $CO_2$ flux may be driven by initialization of the physical (e.g., SST) or*

*biogeochemical (e.g., DIC) ocean state (Li et al., 2016), and that the spatiotemporal coverage of $CO_2$ flux observations is insufficient to fully address predictive skill in our forecast systems.*

L15 Please add Resplandy et al. (2014) which describes how far the decadal variability in ocean carbon fluxes differs between models

We have added a reference to Resplandy et al. (2015) here.

L7-13: Please expand this paragraph, see my major comments

We have expanded the discussion of the initialization procedure in Section 2 (see also our response to your Page 3, line 23 comment):

*CESM-DPLE initializes an ensemble of 40 simulations each year using round-off level (order $10^{-14}$) perturbations in the initial air temperature field (Figure 1). Previous work indicates that this small perturbation in the initial conditions generates a wide divergence in global mean surface temperatures across the ensemble members within about 30 days (V. Yettella, pers. comm., 2018), and the average divergence in globally-integrated, annual-mean forecast $CO_2$ flux across the ensemble members (0.53 Pg C $yr^{-1}$) is an order of magnitude greater than that generated by the preindustrial control simulation of CESM (0.09 Pg C $yr^{-1}$; Lovenduski et al.,2015b).*

L20: a reasonable job is not enough to determine if a model is fitted for purpose. Could you please provide further details such as the spatial correlation, the RMSE : : :

Thank you for catching this! We now include the spatial correlation coefficient for the reconstruction and the observational product (r = 0.79) in the manuscript text:

*In Figure 2, we illustrate the comparison between observationally-based estimates of $CO_2$ flux (from the Landschutzer et al. (2016) $pCO_2$ product) and estimates produced by the reconstruction and coupled CESM-LE over 1982-2015. The model reconstruction does a reasonable job (r = 0.79) of representing observed spatial patterns (in both magnitude and direction) of the flux across most oceanic regions. The globally-integrated air-sea $CO_2$ flux over 1982-2015 from the observational product and model reconstruction are 1.41 and 1.80 Pg C $yr^{-1}$, respectively (directed into the ocean).*

L23: 10-14 Kelvin ? This is a really small perturbation. Have assessed if this initial perturbation lead to populate the full range of model variability as diagnosed from the piControl ? It is important to tell the reader if you chose to populate this uncertainty or instead to stay close from the initial conditions

Indeed, this is a very small perturbation, but work by others suggests that this tiny perturbation generates a wide divergence in global mean surface temperature within approximately 30 days of the initialization (see Figure R3 below). Further, the globally-integrated air-sea $CO_2$ flux from

the preindustrial control run of CESM has a standard deviation of 0.09 PgC yr$^{-1}$, while the average standard deviation (spread across 40 ensemble members) of the forecast air-sea $CO_2$ flux is 0.53 PgC yr$^{-1}$.  We have added a description of these findings to Section 2 of the manuscript:

*CESM-DPLE initializes an ensemble of 40 simulations each year using round-off level (order $10^{-14}$) perturbations in the initial air temperature field (Figure 1).  Previous work indicates that this small perturbation in the initial conditions generates a wide divergence in global mean surface temperatures across the ensemble members within about 30 days (V. Yettella, pers. comm., 2018), and the average divergence in globally-integrated, annual-mean forecast $CO_2$ flux across the ensemble members (0.53 Pg C yr$^{-1}$) is an order of magnitude greater than that generated by the preindustrial control simulation of CESM (0.09 Pg C yr$^{-1}$; Lovenduski et al.,2015b).*

[Figure]

**Figure R3**.  Rapid divergence in global mean surface temperature in the CESM-Large Ensemble, following initialization on January 1, 1920 using order $10^{-14}$ Kelvin perturbations in the initial air temperature field.  Black lines correspond to individual ensemble members, and red line shows ensemble member 1.  Figure from V. Yettella (personal communication, 2018).

L33: Please provide a figure of the ensemble with the drift as in Kim et al 2012 in addition to the de-drifted ensemble. You could add on panel on figure 1 to show that.

We include for the reviewer the companion to Figure 1 that shows the non-drift corrected forecast (see Figure R1 above).  As it is nearly identical to the drift-corrected version, we have elected not to include it in the revised manuscript. Manuscript unchanged in response to comment.

L9-11: I'm a bit puzzled here. Unless I misunderstood McKinley et al 2016 and Lovenduski et al. 2016 used a CMIP5-style CESM and hence CMIP5 forcings to performed their analyses. Here, CESM-DPLE is setup for CMIP6 and hence use CMIP6 forcings, right ? If it does, several external forcing have been revised between CMIP5 and CMIP6. This is the case for the volcanoes (which influence the predictability). Could you please comment this point ?

The CESM-DPLE was generated using the same code base, component model configurations, and historical and projected radiative forcings as in the CESM Large Ensemble (i.e., CMIP5).  To avoid reader confusion, we have removed the reference to the DCPP CMIP6 protocols in Section 2:

*CESM-DPLE consists of a set of initialized, fully-coupled integrations of CESM that adhere to the protocols for Component A of the Decadal Climate Prediction Project (Boer et al., 2016).*

L10-11 What happens if you consider the full observational time-series ? Besides could you please explain why the correlation of the uninitialized simulation slightly increases with lead time ?

We are not considering the observational time-series here, but rather the modeled time-series. In response to Reviewer 2, we have modified the uninitialized simulation correlations to align with the DCPP protocols – it is now the same correlation coefficient for all forecast lead times.

On Figure 3 8 and 9 please indicate the correlation limits (R*) on the graphs and indicates the level of confidence (and the number of degree of freedom used for the t-test) employed; this information is missing.

On Figures 3 and 9 (Figure 8 does not show correlation) we have added asterisks and circles to indicate the maximum lead time in the initialized forecast that is statistically separable from the uninitialized and persistence forecasts.  We perform a Fisher's r to z transformation on the correlation coefficients and compare the resulting z test statistic to the value for the 95% confidence interval (1.96).  We have also modified the text describing Figure 3 to read:

*Figure 3 indicates that the initialized forecast exhibits higher predictability than the uninitialized forecast and the persistence forecast for a lead time of 10 years, though this initialized predictability is only statistically separable from the uninitialized and persistence forecasts for lead years 1-2 and 2, respectively (statistical separation determined via a Fisher's r to z transformation and a comparison of the resulting z test statistic to the value (1.96) for the 95% confidence interval).*

L15: why are you talking about emissions and terrestrial CO2 uptake. Your model set-up employs CO2 concentration as prescribed in the forcing, correct ?

Thanks for catching this – you are correct. We have excised this text.

L25: linearly detrended forecast: have you done this detrended at grill-cell scale or have you applied the same detrending globally ?

The detrending is done on a grid cell by grid cell basis. We have modified the manuscript to better describe this procedure:

*To capture the predictability on interannual timescales, we perform analysis on linearly detrended forecasts in each model grid cell.*

L26: as you suggested that your modelling plateform is able to predict ocean carbon sink up to 7 years in advance it could be usefull to show what happens at lead time greater than 1 year. Could you please replace the figures showing lead-time (LT) 1 by LT 7 and/or moving LT1 Figures as supplemental data

We now include a supplemental figure (Figure S1) showing the predictability for $CO_2$ flux on forecast lead times 2-10 years.

L2: please explain what are your forecast skill and what is the limit for a skillfull predictability at a given confidence

Forecast skill is defined in the previous paragraph as a measure of the ability of the forecast to reproduce the observational record. Since "some skill" is not quantitative, we provide the correlation coefficients in line with the text. Manuscript unchanged in response to comment.

L11: predictability or persistence, please see my major comments

The decision to aggregate the results on the biome scale is not dependent on the drivers of DIC and Alk predictability. Manuscript unchanged in response to comment.

L23: You could state that this biomes does not see an impacts of your initialization procedure. Maybe the sea-ice influence regions are not restored to the observations? This is why I suggest to further develop this point with a new section in the ms, see my major comments

Sea ice is initialized in the same way as the other state variables and thus is not treated differently. As indicated in the following paragraph of the original manuscript, the external forcing dominates the predictability here. Manuscript unchanged in response to comment.

L1: forecast and the uninitialized= remove 'forecast and'

Text excised as suggested.

L15-21: Please further discuss the limit of your approach= for example your initialization procedure fails at capturing the observed variability. You estimate a predictability of 7 years with only 20 years of data (which is not enough). I suggest the authors to discuss this point and hence to highlight the most of the results presented in this work relates to the potential predictability rather than an effective predictability.

In response to this and other comments, we have substantially modified the Conclusions section:

*We analyze output from the CESM-DPLE system to quantify and understand the sources of predictability and predictive skill in global and regional air-sea $CO_2$ flux on annual to decadal timescales.  We find high potential predictability in globally-integrated $CO_2$ flux several years in advance that is engendered by initialization.  This potential predictability is evident across much of the global ocean, driven by predictability in $\Delta pCO_2$ which itself is primarily driven by predictability in surface ocean DIC and Alk.  While the CESM-DPLE system exhibits strong potential predictability, model skill as compared to an observationally-based product remains a challenge to developing useful forecasts.*

*…*

*While the ever-expanding field of decadal climate prediction has the potential to inform policy and management decisions moving forward, decadal forecasts come with several caveats. Initialization shock and drift of the coupled model system, inability of Earth system models to realistically simulate internal variability, uncertain future levels of radiative forcing, and imperfect observations are frequently cited as limitations to making accurate forecasts of the future (Meehl et al., 2014).  In the case of ocean carbon, it is important to note that potential predictability in regional $CO_2$ flux may be driven by initialization of the physical (e.g., SST) or biogeochemical (e.g., DIC) ocean state (Li et al., 2016), and that the spatiotemporal coverage of $CO_2$ flux observations is insufficient to fully address predictive skill in our forecast systems.*

Betts, R. A., Jones, C. D., Knight, J. R., Keeling, R. F., Kennedy, J. J., Wiltshire, A. J., : : : Aragão, L. E. O. C. (2018). A successful prediction of the record CO 2 rise associated with the 2015/2016 El Niño. Philosophical Transactions of the Royal Society B: Biological Sciences, 373(1760), 20170301. https://doi.org/10.1098/rstb.2017.0301
Betts, R. A., Jones, C. D., Knight, J. R., Keeling, R. F., & Kennedy, J. J. (2016). El Niño and a record CO2 rise. Nature Climate Change, 6(9), 806–810. https://doi.org/10.1038/nclimate3063 Keenlyside, N. S., Latif, M., Jungclaus, J., Kornblueh, L., & Roeckner, E. (2008). Advancing decadal-scale climate prediction in the North Atlantic sector. Nature, 453(7191), 84–88. https://doi.org/10.1038/nature06921
Kim, H. M., Webster, P. J., & Curry, J. A. (2012). Evaluation of shortterm climate change prediction in multi-model CMIP5 decadal hindcasts. Geophysical Research Letters, 39(10), L10701. https://doi.org/10.1029/2012GL051644

Li, H., Ilyina, T., Müller, W. A., & Sienz, F. (2016). Decadal predictions of the North Atlantic CO2 uptake. Nature Communications, 7(May 2015), 11076. https://doi.org/10.1038/ncomms11076 Resplandy, L., Séférian, R., & Bopp, L. (2015). Natural variability of $CO_2$ and $O_2$ fluxes: What can we learn from centuries-long climate models simulations? Journal of Geophysical Research: Oceans, 120(1). https://doi.org/10.1002/2014JC010463 Séférian, R., Berthet, S., & Chevallier, M. (2018). Assessing the Decadal Predictability of Land and Ocean Carbon Uptake. Geophysical Research Letters, 45(5), 2455–2466. https://doi.org/10.1002/2017GL076092 Séférian, R., Bopp, L., Gehlen, M., Swingedouw, D., Mignot, J., Guilyardi, E., & Servonnat, J. (2014). Multiyear predictability of tropical marine productivity. Proceedings of the National Academy of Sciences of the United States of America, 111(32). https://doi.org/10.1073/pnas.1315855111 Smith, D. M., Cusack, S., Colman, A. W., Folland, C. K., Harris, G. R., & Murphy, J. M. (2007). Improved Surface Temperature Prediction for the Coming Decade from a Global Climate Model. Science, 317(5839), 796–799. https://doi.org/10.1126/science.1139540

**Anonymous Referee #2**

The authors have investigated the predictability and predictive skill of the ocean carbon uptake by using a large ensemble of 40-member decadal prediction and historical simulations based on NCAR CESM. They found a prominent improved predictability of the ocean carbon uptake in the initialized simulations in comparing with the uninitialized historical simulations and the persistence forecast. Furthermore, they attribute the predictability of ocean carbon uptake to the dissolved inorganic carbon and alkalinity. The outcome of this study is an important contribution for understanding and predicting variations of the ocean carbon uptake and the global carbon cycle, which are crucial for estimating climate change. Moreover, reconstruction and near-term predictions of global carbon cycle show large potential for supporting the future carbon stocktaking. Therefore the study on this topic merits publication on the Earth System Dynamics.

The manuscript is well written and the results are clearly stated, however, the conclusions are not quantitatively precise and not statistically robust from the results. This together with some other issues listed below prevents me accepting this manuscript at its present format.

1. The authors claimed a potential predictive skill of up to 7 years in the abstract and conclusions. Is this conclusion from Fig. 3d by comparing the initialized forecast to the uninitialized forecast? The authors did not do a statistical test if the difference between initialized and the uninitialized forecast is significant. The red circles only show if the correlation of the initialized forecast itself is significant. It seems to me that the initialized forecast (red dots) at lead time of 6 and 7 years are very close to the uninitialized forecast, these are probably not significantly distinguishable. As the improved skill due to initialization is a main quantitative conclusion in this manuscript, it requires a sound significant test, such as the commonly used bootstrap method (Goddard et al., 2013), which is also suggested by the Decadal Climate Prediction Project (DCPP) (Boer et al., 2016). In addition, the authors only show time-series and maps of predictability at lead time of 1 year. Given the high predictability of ocean carbon uptake as stated in this study, time-series and maps of predictability at longer lead time at least of 2 years are more representative.

Thank you for bringing this to our attention.  We have revisited the statistical framework for the correlations and now include a robust assessment of the statistical separation of the anomaly correlation coefficients.  To do this, we perform a Fisher's r to z transformation on the correlation coefficients and compare the resulting z test statistic to the value for the 95% confidence interval (1.96).  We have modified Figures 3, 9, and 10 and Table 1 to reflect this change, and have modified the text in the abstract, conclusions, and throughout the main manuscript to indicate the findings of this statistical test.

2. The authors estimated both potential predictability against reconstruction and predictive skill against observation-based data product. The two results were separately discussed in the main text, however, the conclusions are mixed especially in the abstract. It's quite difficult for

the readers to distinguish the origin of the conclusions, they are from potential skill or skill against observation. For this reason, the abstract needs to be reorganized and make it clearer. Furthermore, the connections between potential predictabiltiy and predictive skill are weak in the manuscript. How consistent/inconsistent are the predictability and the predictive skill? What would be the implication of potential predictability to the predictive skill versus observation?

Thank you for this suggestion. We have reorganized the abstract to more clearly state the difference between the potential to predict and the actual skill:

*Interannual variations in air-sea fluxes of carbon dioxide ($CO_2$) impact the global carbon cycle and climate system, and previous studies suggest that these variations may be predictable in the near-term (from a year to a decade in advance). Here, we quantify and understand the sources of near-term predictability and predictive skill in air -sea $CO_2$ flux on global and regional scales by analyzing output from a novel set of retrospective decadal forecasts of an Earth system model. These forecasts exhibit the potential to predict year-to-year variations in the globally-integrated air-sea $CO_2$ flux several years in advance, as indicated by the high correlation of the forecasts with a model reconstruction of past $CO_2$ flux evolution. This potential predictability exceeds that obtained solely from foreknowledge of variations in external forcing or a simple persistence forecast, with the longest-lasting forecast enhancement in the subantarctic Southern Ocean and the northern North Atlantic. Potential predictability in $CO_2$ flux variations are largely driven by predictability in the surface ocean partial pressure of $CO_2$, which itself is a function of predictability in surface ocean dissolved inorganic carbon and alkalinity. The potential predictability, however, is not realized as predictive skill, as indicated by the moderate to low correlation of the forecasts with an observationally-based $CO_2$ flux product. Nevertheless, our results suggest that year-to-year variations in ocean carbon uptake have the potential to be predicted well in advance, and establish a precedent for forecasting air-sea $CO_2$ flux in the near future.*

While we maintain the separation of potential predictability and the predictive skill in the manuscript (for reader clarity), we have rewritten the first paragraph of the conclusions to better integrate the two:

*We analyze output from the CESM-DPLE system to quantify and understand the sources of predictability and predictive skill in global and regional air-sea $CO_2$ flux on annual to decadal timescales. We find high potential predictability in globally-integrated $CO_2$ flux several years in advance that is engendered by initialization. This potential predictability is evident across much of the global ocean, driven by predictability in $\Delta pCO_2$, which itself is primarily driven by predictability in surface ocean DIC and Alk. While the CESM-DPLE system exhibits strong potential predictability, model skill as compared to an observationally-based product remains a challenge to developing useful forecasts.*

3. The initialized simulations were started from a forced ocean-sea ice simulation for the ocean-sea ice component, but were started from the CESM Large Ensemble for the atmosphere and

the land components (details were described in page 3 lines 8-13). This means that the ocean and the atmosphere and land are most probably in different climate state/phase, they need to adjust to each other and approach a new equalibrium. The mismatch of initial conditions in the ocean and in the atmosphere and land would affect the variations and predictions of the system, especially for the carbon flux across the boundaries. Discussions of the effects of mismatch in the ocean and the atmophere and land are necessary. Can the model drift due to the mismatch be largely eliminated by the drift correction?

Indeed, the full-field initialization procedure generates a drift in the forecasts. As stated in the original manuscript, we correct this drift by transforming to anomalies from a drifting climatology. The drifting climatology is statistically robust due to the large number (n=40) of ensemble members in each forecast and the large number (n=62) of start years. We include for the reviewer the companion to Figure 1 that shows the non-drift corrected forecast (see Figure R1 above). It is nearly identical to the original Figure 1, indicating that the drift in $CO_2$ flux is not particularly large, and thus further description of the drift is unnecessary.

4. As stated in McKinley et al. (2016), some ensemble members of the CESM-LE have problem in the ocean biogeochemical outputs. McKinley et al. (2016) used only 32 ensemble members of the CESM-LE, because some ensemble members were discarded due to a setup error which leads to corrupts of ocean biogeochemical output. In this study, the authors use 40 ensemble members as written in Page 4 lines 5-9. How do the authors treat the ensemble members with setup error in this study?

Thank you for catching this! We have excluded the LE ensemble members that have corrupted biogeochemistry. We have modified the text to reflect this.

5. The numbers in Fig. 10 are not significant and deducible from Fig. 9. For instance, the maximum forecast lead time in biome 3 (NP STSS) is 8 years in Fig. 10, but if we look at Fig. 9a, the correlations at lead time beyond 4 years are not significant and end up with less than 0.2 at lead time of 8 years. As for biome 4 (NP STPS), the maximum forecast lead time is 7 years in Fig. 10, but the initialized forecast skill is not significantly higher than the uninitialized forecast skill at lead time of 5 years in Fig. 9b. Therefore, I think the numbers in Fig. 10 need to be carefully checked by taking into account the significant test and the relative magnitude of the correlations.

We have added hatching to Figure 10 to indicate the maximum lead time in the initialized forecast that is statistically separable from the uninitialized and persistence forecasts. To do this, we perform a Fisher's r to z transformation on the correlation coefficients and compare the resulting z test statistic to the value for the 95% confidence interval (1.96). As suggested by Reviewer 1, we now include a supplemental figure (Figure S1) that shows maps of predictability for forecast lead years 2-10.

6. Table 1: the table caption and the title of the columns are unclear. I guess the "Initialized

forecast" and the "uninitialized forecast" refer to forecast skill versus reconstruction, and the "Forecast skill" refer to forecast skill versus observation-based products. The time period used to calculate the correlations needs to be specified, especially for the "Forecast skill" which use much shorter period. In addition, statistical significant test information by highlighting of the numbers will be also helpful. Moreover, a table of predictability for the maximum forecast lead time will be necessary as supplementary information to Fig. 10.

We have clarified the table column headings with footnotes, and added a column that reports the maximum forecast lead time and the maximum statistically separable lead time. As suggested by Reviewer 1, we now include a supplemental figure (Figure S1) that shows maps of predictability for forecast lead years 2-10.

7. Fig. 2: how different is the reconstruction comparing to the uninitialized simulations? Is the reconstruction closer to observations than the uninitialized simulaitons? It would be more informative to also include the climatology of the uninitialized simulations.

We have added a panel to Figure 2 that shows the climatology from the uninitialized CESM-LE simulations. We note very minor differences between the two.

8. It is not introduced but I guess the authors use different time period for the drift correction and correlation calculation along different lead time. As shown in Fig. 3d, the red dashed line has a slightly positive trend, which indicates that the authors use different time period for the correlation calculation for different lead time. To make a consistent estimate of predictive skill along all the forecast range, it is better to use the same time period for all the lead years as suggested by DCPP (Boer et al., 2016, Appendix E) and previous studies focusing on the physical predictions (Hawkins et al., 2014; Smith et al., 2013).

Thank you for bringing this to our attention. We now use a consistent time period for all of the correlation calculations from the uninitialized simulation, and have updated Figures 3, 9, and 10 and Table 1 accordingly.

9. Page 3 line 7: are the historical external forcings from CMIP5 or CMIP6 (i.e., the 5$^{th}$ of 6th Coupled Model Intercomparison Project)?

The CESM-DPLE was generated using the same code base, component model configurations, and historical and projected radiative forcings as in the CESM Large Ensemble (i.e., CMIP5). To avoid reader confusion, we have removed the reference to the DCPP CMIP6 protocols in Section 2:

*CESM-DPLE consists of a set of initialized, fully-coupled integrations of CESM that adhere to the protocols for Component A of the Decadal Climate Prediction Project (Boer et al., 2016).*

10. Page 5 line 16: ": : :for those forecasting year-to-year changes: : :" should be ": : :for those reproducing year-to-year changes: : :"

The Global Carbon Project forecasts ocean carbon uptake for the current year before it is measured/quantified, and we intended to draw attention to this in the text.  Manuscript unchanged in response to comment.

11. Page 6 line 5: ": : :the anomaly correlation coefficients are scaled to CO2 flux units: : :" The correlation coefficient itself is uniformed and has no unit, there is no need to further scale it. What are the results based on the correlation coefficients without the scaling? I think the results without scaling are similar to those based on the scaled correlations. It worths to check. One more question on the scaling formular: how do the authors calculate the @_/@x, how long is the time step?

We include for the reviewer a figure (Figure R4 below) that shows the predictability/correlation coefficients before applying the scaling.  This figure suggests a more important role for solubility than the scaled version, and gives the reader a false impression of the role of solubility in the predictability of $CO_2$ flux.  We have opted to maintain the scaled version in the manuscript.

With regard to the scaling formula: the timestep is annual-mean.  We now indicate this in the manuscript text.

[Figure]

**Figure R4**. Predictability in the drivers of air-sea $CO_2$ flux during forecast year 1, as indicated by the correlation of forecast and reconstruction anomalies in the (a) gas-exchange coefficient, (b) solubility, (c) sea ice fraction, and (d) $\Delta pCO_2$.  Correlation coefficients that are not statistically significant at the 95% level using a t test are assigned a value of zeros.

12. Page 6 line 22-23: "The similar predictability of DIC and Alk across many regions hints at an important role for ocean circulation, rather than biological productivity: : :, in CO2 flux predictability." From this I understand that the biological productivity is a secondary regulation

of CO2 flux, therefore the biome division is probably not a proper way to divide the global ocean for CO2 flux predictions. The last sentence is the same as line 8-9 on Page 7.

We have overlain a map of the biome boundaries in Figure 2a to illustrate that these boundaries appropriately capture differences in mean air-sea $CO_2$ flux.

13. Page 8 line 26: "Li and Ilyina (2018)" should be "Li et al. (2016)", right?

Good catch!  We now make reference to Li et al. (2016).

14. Figure 4 caption: "CESM-DPLE initialized forecast lead year 1" needs to be revised and includes information of the counterpart of the correlation, e.g., "CESM-DPLE initialized forecast for lead year 1 with the reconstruction".

We have modified the caption as suggested.

15. Figure 9: are the correlations based on detrended time series?

Yes.  We have modified the caption to reflect this.

References: Boer, G. J., et al.: The Decadal Climate Prediction Project (DCPP) contribution to CMIP6, Geosci. Model Dev., 9, 3751-3777, https://doi.org/10.5194/gmd-9-3751-2016, 2016. Goddard, L., et al.: A verification framework for interannual to decadal predictions experiments, Clim. Dynam., 40, 245–272, doi:10.1007/s00382-012-1481-2, 2013. Hawkins, E., Dong, B., Robson, J., Sutton, R., and Smith, D.: The interpretation and use of biases in decadal climate predictions, J. Climate, 27, 2931–2947, doi:10.1175/JCLI-D-13-00473.1, 2014. McKinley, G. A., Pilcher, D. J., Fay, A. R., Lindsay, K., Long, M. C., and Lovenduski, N. S.: Timescales for detection of trends in the ocean carbon sink, Nature, 530, 469–472, http://dx.doi.org/10.1038/nature16958, 2016. Smith, D. M., Eade, R., and Pohlmann, H.: A comparison of fullfield and anomaly initialization for seasonal to decadal climate prediction, Clim. Dynam., 41, 3325–3338, doi:10.1007/s00382-013-1683-2, 2013.

[revised manuscript text omitted]